# Optimization of Extraction Process and the Antioxidant Activity of Phenolics from *Sanghuangporus baumii*

**DOI:** 10.3390/molecules26133850

**Published:** 2021-06-24

**Authors:** Na Zheng, Yongfei Ming, Jianzhi Chu, Shude Yang, Guochao Wu, Weihuan Li, Rui Zhang, Xianhao Cheng

**Affiliations:** 1Shandong Key Lab of Edible Mushroom Technology, School of Agriculture, Ludong University, Yantai 264025, China; zn1221914@163.com (N.Z.); chu_jianzhi@163.com (J.C.); sdyang68@126.com (S.Y.); wuguochaoo@163.com (G.W.); lwh1979@163.com (W.L.); 2School of Life Science, Ludong University, Yantai 264025, China; mingyf@163.com

**Keywords:** polyphenols, *Sanghuangporus baumii*, deep eutectic solvent, response surface methodology, antioxidant

## Abstract

*Sanghuangporus baumii*, is a widely used medicinal fungus. The polyphenols extracted from this fungus exert antioxidant, anti-inflammatory, and hypoglycemic effects. In this study, polyphenols from the fruiting bodies of *S. baumii* were obtained using the deep eutectic solvent (DES) extraction method. The factors affecting the extraction yield were investigated at different conditions. Based on the results from single-factor experiments, response surface methodology was used to optimize the extraction conditions. The scavenging ability of the polyphenols on •OH, DPPH, and ABTS^+^ was determined. The results showed that the DES system composed of choline chloride and malic acid had the best extraction yield (6.37 mg/g). The optimal extraction parameters for response surface methodology were as follows: 42 min, 58 ℃, 1:34 solid–liquid (mg/mL), and water content of 39%. Under these conditions, the yield of polyphenols was the highest (12.58 mg/g). At 0.30 mg/mL, the scavenging ability of the polyphenols on •OH, DPPH, and ABTS^+^ was 95.71%, 91.08%, and 85.52%, respectively. Thus, the method using DES was more effective than the conventional method of extracting phenolic compounds from the fruiting bodies of *S. baumii*. Moreover, the extracted polyphenols exhibited potent antioxidant activity.

## 1. Introduction

Sanghuang is a popular medicinal fungus available in China, with a reputation of being “forest gold.” Sanghuang has been used as traditional Chinese medicine for more than 2000 years and is recorded in many ancient Chinese books [1]. According to the Shennong Materia Medica Classic and Compendium of Materia Medica, its main effects are as following: relieving diarrhea, breaking and leaking down the belt, and relieving diarrhea by spleen deficiency [2]. Although ancient Chinese books indicate the use of Sanghuang as medicine, the species of Sanghuang that can be used cannot be identified solely based on their description in the literature, because of the existence of several varieties of Sanghuang. In recent years, scholars still have differences in opinion regarding the classification of Sanghuang. However, studies show that almost all varieties of Sanghuang exert several pharmacological effects.

In 2012, a new species was identified and named *Inonotus Sanghuang* Sheng H. Wu, T. Hatt. & Y.C. Dai [3]. In 2016, Wu Shenghua et al. combined molecular biological identification and morphological analysis, and the genus *Sanghuangporus* Sheng H. Wu, L.W. Zhou & Y.C. Dai were established. Currently, there are 14 species in this genus, of which nine are found in China [4]. Additionally, through the textual research of Chinese herbology studies, Bao Haiying et al. believe that *Inonotus hispidus* (Bull.) P. Karst. is the closest to Sanghuang. At the same time, *Phelinus* Quel., *Enchir. Fung*., and *Inonotus* P. Karst, which are close to the genus *Sanghuangporus,* can also be called Sanghuang [5]. To date, researchers continue to actively explore the classification and uses of Sanghuang. *Sanghuangporus Baumii* (Pilat) L.W. Zhou & Y.C. Dai is a fungus belonging to the genus *Sanghuangporus*. Compared with *S. Sanghuang* Sheng H. Wu, T. Hatt. & Y.C. Dai, *S. baumii* is known for its ability to grow easily; it has a short growth cycle and large-scale cultivation can be achieved.

Sanghuang contains several pharmacologically active compounds, including polysaccharides, flavonoids, pyrone, polyphenols, terpenoids, and sterols, which have been reported to cure diseases [6,7,8,9]. To date, the main polyphenols reported in Sanghuang include phellibaumin A, phelligridin C, phelligridin D, hypholomin B, protocatechualdehyde, interfungin B, and chlorogenic acid [10,11,12,13,14]. Pharmacological studies have shown that polyphenols are associated with improving immunity and exert antitumor, antioxidant, antihyperglycemic, and hepatoprotective effects [15,16,17,18]. However, in recent years, researches on Sanghuang is mainly focused on polysaccharides and flavonoids, and very few studies have studied the polyphenols present in this fungus. Therefore, in-depth research on the extraction and pharmacological activity of the polyphenols of Sanghuang is well warranted.

Ethanol, methanol, and petroleum ether are some of the organic solvents commonly used for the extraction and separation of polyphenols from natural sources [19]. Other important extraction methods include ultrasound-assisted ethanol extraction [20], microwave-assisted methanol extraction [21], enzyme-assisted extraction [22], supercritical extraction [23], and subcritical water extraction [24]. These traditional methods have some disadvantages, such as extended extraction time, low purity of the extracted components, low extraction yield, and the possibility of environmental pollution. Therefore, it is necessary to identify new and green extraction solvents and design simple extraction schemes with high efficiency to replace some of the commonly used extraction solvents.

Deep eutectic solvents (DES) are green extraction solvents that can replace organic solvents. This system comprises a transparent solvent that is obtained by mixing two or more substances in a certain proportion. Abbott et al. [25] found that amides and quaternary ammonium salts mixed in a certain molar ratio can form low-melting eutectic mixtures having special solvent properties and named them DES. DES can be used for simple synthetic processes; moreover, they are inexpensive, non-toxic, low volatility, high solubility, nonflammability, environmentally friendly, reusable, and degradable [26], all of which conform to the principles and concepts of green chemistry. Owing to these unique properties, DES has attracted increasing attention from scholars and has been widely used in catalysis, organic synthesis, and analytical chemistry [27,28,29]. In this study, we selected a suitable eutectic solvent from among many DES to extract polyphenols from *S. baumii.* We optimized the extraction using a single-factor test combined with the response surface curve method. In addition, we performed in vitro experiments to determine the antioxidant activities of the extracted polyphenols. The findings of our study provide the corresponding theoretical basis for further research, including the development and utilization of Sanghuang in health products.

## 2. Results

### 2.1. Effect of Different Eutectic Solvents on the Extraction of S. baumii Polyphenols

The extraction yields of polyphenols varied to some extent because the hydrogen donor for each combination of DES differed. To determine the optimal composition of DES to extract *S. baumii* polyphenols, a total of 8 DES combinations were selected in this experiment. At the same time, different concentrations of ethanol were selected as controls to more directly reflect the differences between the two extraction systems. First, using a solid–liquid ratio of 50 mg/mL, ultrasonic-assisted extraction time of 30 min, a temperature of 50 ℃, and an extraction power of 200 W, 40 kHz, the effects of 50%, 60%, 70%, 80% and 90% ethanol on the yields of *S. baumii* polyphenols were compared. The experimental results are shown in Figure 1. We found that the highest yield of phenol was 4.20 mg/g when 60% ethanol was used as the extraction solvent.

Eight different DES with 40% water content were used to extract *S. baumii* polyphenols using a solid–liquid ratio of 50 mg/mL, ultrasonic-assisted extraction for 30 min, temperature of 50 ℃, and extraction power of 200 W, 40 kHz; 60% ethanol was used as a control. The results are shown in Figure 2. With the exception of the choline chloride and glycerol (DES-3) system, the extraction yield of other combinations of DES was better than that obtained using ethanol. Other combinations of DES led to yields higher than those obtained using ethanol extraction. The choline chloride and malic acid (DES-1) combination for the extraction of polyphenols led to the highest yield of 6.37 mg/g. The extraction yield of choline chloride and malic acid DES was 1.5 times that of 60% ethanol. Therefore, DES can be used to replace organic solvents and show remarkable application potential in the extraction of polyphenols. Based on our findings, DES-1 was selected as the extraction solvent for subsequent experiments.

### 2.2. Results from Single-Factor Experiments

The results of the single factor test are shown in Figure 3. The experimental results show that when the water content was 40%, the yield of the extracted polyphenols reached the maximum, shown in Figure 3A. When the extraction time was 40 min, the polyphenols yield was not significantly different from that obtained using an extraction time of 50 min. From the view of energy and cost savings, 40 min was selected as the optimal extraction time of *S. baumii* polyphenols, shown in Figure 3B. The polyphenols yield was the highest at 60 ℃, shown in Figure 3C. The polyphenols yield was the highest at a ratio of 40 mg/mL, shown in Figure 3D.

### 2.3. Optimization of Polyphenols Extraction Using Response Surface Methodology

The design plan and results obtained using Design-Expert 12 are shown in Table 1. Multiple linear regression analysis was conducted based on the results in Table 1, and the linear regression equation of the experiment was conducted as follows:$$y=-93.46+0.81A+0.69B+2.19C+0.72D+0.015\mathrm{AB}\;-\;0.007\mathrm{AC}\;-\;0.004\mathrm{AD}\;-0.005BD+0.002CD-0.05A^{2}-0.006B^{2}-0.018C^{2}-0.006D^{2}$$

Results of the regression equation are shown in Table 2. From the equation, the correlation coefficient was determined to be R^2^ = 0.9271. For the regression model, *p* < 0.0001 and reached an extremely significant level; the loss of fit term was 0.0716, *p* > 0.05. The loss of fit term is not significant, indicating that the selection of the model was reasonable and feasible, and could be used to analyze and predict the extraction of polyphenols. In the regression equation, *p* values of A and C were <0.05, indicating that the two influencing factors of A and C had significant effects on the extraction yield of polyphenols. At the same time, through the comparison of the value of *F*, we can judge the influence of various factors on the extraction yield of *S. baumii*. The influence of each factor can be presented as follows: C > A > B > D.

A 3D response surface map was used to represent the interaction between various factors, shown in Figure 4. The interactions among various factors can be seen more intuitively from the 3D figure. The surfaces of factors A and C are relatively steep, while the changes in factors B and D are relatively gentle, indicating that A and C have a greater impact on the extraction yield of polyphenols.

Four factors were analyzed using the Desk-Expert12, and the optimal extraction conditions were determined to be as follows: water content was 38.76%, extraction time was 42.45 min, extraction temperature was 57.72 °C, and solid–liquid ratio was 34.35 mg/mL. Using these optimal conditions, the predicted extraction yield of polyphenols was 12.48 mg/g. To ensure the experimental feasibility, the optimal extraction conditions were modified as follows: water content was 39%, extraction time was 42 min, extraction temperature was 58 ℃, and the solid–liquid ratio was 34 mg/mL. Using these modified conditions, each experiment was performed in triplicate. The extraction yield of *S. baumii* polyphenols was 12.58 mg/g and the relative deviation was 0.85%, both of which were consistent with the corresponding theoretical values. These findings indicated that our model was reasonable and feasible and could be used to predict and analyze the extraction yield of *S. baumii* polyphenols. Moreover, the efficiency of DES to extract polyphenols were more than two times higher (12.58 mg/g) than that obtained using the conventional method with ethanol (4.20 mg/g). These findings indicated the obvious superiority of our method.

### 2.4. Antioxidant Activity of S. baumii Polyphenols

The experimental results of the antioxidant activity of polyphenols were shown in Figure 5.

The •OH-scavenging activity of *S. baumii* polyphenols is higher compared with that of VC (Figure 5A). The free-radical scavenging ability of the polyphenols increased with an increase in the concentration of the extracted polyphenols. The scavenging ability was as high as 95.71% when the polyphenols concentration was 0.30 mg/mL. The scavenging ability was comparable to, or even higher than, VC at low concentrations.

The DPPH scavenging rate of *S. baumii* polyphenols is shown as (Figure 5B). VC was used as a control. We found an increase in the rate of DPPH scavenging with increasing polyphenols concentration. When the polyphenols concentration was 0.30 mg/mL, the DPPH scavenging ability was the highest and reached 91.08%, which was similar to the scavenging rate by VC.

Results of ABTS^+^ scavenging rate by *S. baumii* polyphenols are shown as (Figure 5C). VC was used as a control. We found that ABTS^+^ scavenging by *S. baumii* polyphenols increased with increasing polyphenols concentration. The ABTS^+^-scavenging ability was the highest at 85.52% when the polyphenols concentration was 0.30 mg/mL.

## 3. Discussion

Currently, studies on the active compounds of Sanghuang are mostly focused on polysaccharides, and in-depth studies on polyphenols are still limited. In this study, DES extraction was used to greatly improve the polyphenol yields of *S. baumii*. Although DES can better dissolve polyphenols, their solubility and extraction yields will differ based on the DES used.

For this study, we selected eight types of DESs and found that the one composed of choline chloride, and malic acid had the highest extraction yield. The polarity of the DES composed of choline chloride and malic acid is higher than that of the ethanol system. The types of polyphenols in Sanghuang are complicated. Based on our separation and analysis experiments, Sanghung is rich in some polyphenols with strong polarities, such as gallic acid, phellibaumin A, chlorogenic acid, etc. These polyphenols have strong hydrogen bond interaction with DES and can disperse into DES quickly. Therefore, The DES greatly improves the yield of polyphenols with a higher polarity in Sanghuang.

In the single factor experiments, the extraction yield of polyphenols first increased, and then decreased with an increase in water content. This was likely because when the water content was low, the viscosity of DES was high, which hindered the interaction between *S. baumii* and DES, thereby reducing the extraction rate. When the water content was very high, the interaction force between *S. baumii* and DES decreased, resulting in a lower extraction yield. With an increase in extraction time, the extraction rate of polyphenols increased initially but then tended to stabilize. This pattern could be explained on the basis that ultrasound treatment damaged the cell walls and promoted the dissolution of intracellular polyphenols as the time was increased. After reaching the maximum value, no further increase was observed. With an increase in temperature, the extraction rate of polyphenols increased initially and then decreased. This finding could be attributed to the increase in temperature resulting in the destruction of the cell wall, leading to the dissolution of the intracellular polyphenols. However, with this increase in temperature, some unstable polyphenolic components were destroyed, and other substances were generated, which led to a decrease in the rate of extraction. With an increase in the solid–liquid ratio, the extraction rate of polyphenols showed a trend of an initial increase, followed by the attainment of stability. This may be because the solubility property of DES did not increase after reaching saturation. In terms of pharmacological activity, our in vitro antioxidant studies suggested that the mulberry polyphenols of *S. baumii* had good antioxidant activity. Although we made some interesting discoveries, our study has some limitations. For example, the polyphenols used in the antioxidant experiments were crude compounds obtained after preliminary purification; moreover, the reusability of DES has yet to be proven. The complicated purification procedure of polyphenols is another problem that should be addressed in subsequent experiments.

In future studies, we will continue to further purify the monomer polyphenols. Additionally, we intend to perform more in vivo and in vitro experimental studies to further explore the anti-inflammatory, hypoglycemic, and antitumor effects of the polyphenols of mulberry flavors.

## 4. Materials and Methods

### 4.1. Materials, Reagents, and Instruments

Materials: *S. baumii* was cultured for six months in our laboratory and used in this study. DNA kits were used for ITS sequence amplification and sent to Huada Gene Sequencing Company (Qingdao, China). The measured gene sequences were deposited in the GenBank database to Blast the sequence alignment and combined with the morphological identification, *S. baumii* was identified. The fruiting bodies were dried at 50–60 °C, pulverized in a grinder, passed through a 40-mesh sieve(The aperture is 0.425 nm), and stored at 4 ℃ until further use.

Reagents: Choline chloride, lactic acid, glycerol, malonic acid, urea, glycol, 1,4-butanediol, glycerol, potassium persulfate, trichloroacetic acid, ferric chloride, potassium ferricyanide, salicylic acid, sodium carbonate, ferrous sulfate, vitamin C (VC) were of analytical grade and obtained from Sinopill Chemical Reagents Co., Ltd. (Tianjin, China). Gallic acid monohydrate standard (purity > 98%) analytical pure was purchased from Cool Chemical Technology Co., Ltd. (Beijing, China). Folin was obtained from Shanghai Jinsui Biotechnology Co., Ltd. (Shanghai, China). and analytical grade hydrogen peroxide (H_2_O_2_) was procured from Tianjin Ruijin Chemical Co., Ltd. (Tianjin, China). 1,1-diphenyl-2-picrylhydrazyl (DPPH), 2,2’-azino-bis(3-ethylbenzothiazoline-6-sulfonic acid) (ABTS^+^) was purchased from Shanghai Maclin Biochemical Technology Co., Ltd. (Shanghai, China).

Experimental apparatus: A 2500C grinder (Yongkang Hongsun Electromechanical Co., Ltd., Zhejiang, China), SQP electronic balance (Sartorius Scientific Instruments, Beijing, China), KQ-500DE ultrasonic extractor (Kunshan Ultrasonic Instrument Co., Ltd. Jiangsu, China), HHS-21-6 thermostat water bath (Shanghai Boxun Industrial Co., Ltd., Medical Equipment Factory, Shanghai, China), T6 spectrophotometer (Beijing Purkay General Instrument Co. Ltd., Beijing, China), NEO15R high-speed refrigerated centrifuge (Shanghai Lishen Scientific Instrument Co., Ltd., Shanghai, China), and DHT-450A high-temperature blast drying oven (Shanghai Daohan Industrial Co., Ltd., Shanghai, China) were the instruments used in this study.

### 4.2. Method

#### 4.2.1. Preparation of DES

DES is composed of two parts: the hydrogen donor (choline chloride) and hydrogen acceptor (malic acid, lactic acid, etc.). The two parts were weighed in a certain molar ratio, placed in a round-bottomed flask, and heated in a water bath at 80–90 ℃ for 4–6 h with constant stirring until a colorless transparent liquid was obtained. The hydrogen bonding force between the compounds causes the high viscosity of DES. The addition of a certain volume of water can reduce viscosity and enable the sample to effectively combine with the eutectic solvent. In this study, different eutectic solvents were synthesized as listed in Table 3.

#### 4.2.2. Construction of the Standard Curve

Polyphenol content was determined using Folinol colorimetry [30] with some modifications. Gallic acid monohydrate was used as a standard; 0.1000 g of gallic acid was accurately weighed, dissolved in distilled water, and made up to a volume of 100 mL in a volumetric flask to obtain a 1 mg/mL standard solution. The standard solution (0, 1, 2, 3, 4, and 5 mL) was each pipetted into a 100-mL volumetric flask and the volume was made up with distilled water. The mass concentrations of the prepared solutions were 0, 10, 20, 30, 40, and 50 mg/L, respectively. The sample (1 mL) was taken in a 10-mL volumetric flask and 1 mL of Folinol reagent was added. The mixture was allowed to stand for 5 min, following which 3 mL of 7.5% Na_2_CO_3_ was added and the volume was made up to 10 mL with distilled water. Three replicates were performed for each dilution. The samples were mixed well and left undisturbed in the dark for 2 h. Lastly, the absorbance was determined at 765 nm using appropriate blanks. The mass concentration of gallic acid was plotted as the abaxial coordinate and the absorbance as the vertical coordinate to obtain the standard curve. The regression equation was determined to be y = 0.0127x + 0.0001 (R^2^ = 0.9998).

#### 4.2.3. Determination of Polyphenols Yield

The fruiting body powder of *S. baumii* was weighed and a proportional volume of DES was added. The sample was subjected to ultrasonic-assisted extraction in accordance with the different extraction conditions. Next, the sonicated samples were centrifuged at 5000 rpm for 10 min. The supernatant obtained after centrifugation was diluted according to the method stated in Section 4.2.2 and the polyphenol concentration was determined. Polyphenols content was calculated using the following Equation (1):(1)Polyphenol content (mg/g))=(C*V*n)/m
where C is the mass concentration of the extract (mg/mL), V is the volume of the solution (mL), n is the dilution factor, and m is the dry weight of the sample (g).

### 4.3. Single-Factor Experiment of Ultrasonic-Assisted Extraction of Polyphenols from S. baumii Using DES

#### 4.3.1. Effect of Water Content on the Extraction Yield of Polyphenols from *S. baumii* Using DES

The fruiting-body powder of *S. baumii* was dissolved to yield a 50 mg/mL solution. DES with a water content of 20%, 30%, 40%, 50%, and 60% were used to extract the samples in an ultrasonicator operated with an extraction power of 200 W, 40 kHz, extraction temperature of 50 ℃, and extraction time of 30 min, 5 mL were extracted respectively. Each experiment was performed in triplicate. The optimal water content of DES was determined by comparing the amount of phenolic content obtained from each extraction.

#### 4.3.2. Effect of Extraction Time on Polyphenol Yield from *S. baumii*

The fruiting-body powder of *S. baumii* was dissolved to yield a 50 mg/mL solution. DES with 40% water content were used to extract the samples in the ultrasonic extractor with an extraction power of 200 W, 40 kHz, extraction temperature of 50 ℃, and extraction time of 20, 30, 40, 50, and 60 min, 5mL were extracted respectively. Each experiment was performed in triplicate. The optimal extraction time was determined by comparing the number of polyphenols extracted at the end of each time point.

#### 4.3.3. Effect of Extraction Temperature on the Extraction Yield of Polyphenols from *S. baumii*

The fruiting-body powder of *S. baumii* was dissolved to yield a 50 mg/mL solution. DES with 40% water content were used to extract the samples in an ultrasonic extractor with an extraction power of 200 W, 40 kHz, extraction time of 30 min and extraction temperature of 40 ℃, 50 ℃, 60 ℃, 70 ℃, and 80 ℃, 5mL were extracted respectively. Each experiment was performed in triplicate. The optimal extraction temperature was determined by comparing the number of polyphenols from each extraction.

#### 4.3.4. Effect of Solid-Liquid Ratio on the Extraction Yield of Polyphenols from *S. baumii*

The fruiting body powder of *S. baumii* was dissolved to yield concentrations of 30, 40, 50, 60, and 70 mg/mL. DES with 40% water content were used to extract samples in an ultrasonic extractor with an extraction power of 200 W, 40 kHz, extraction temperature of 50 ℃, and extraction time of 30 min, 5mL were extracted respectively. To ensure accuracy, each experiment was performed in triplicate. The optimal solid-liquid ratio was determined by comparing the number of polyphenols obtained from each extraction.

### 4.4. Optimization of the Extraction Process of Polyphenols Using Response Surface Methodology

According to the results of the single-factor experiment and the experimental principle of Box-Behnken in the response surface curve method, response surface analysis was performed with four factors and three levels using Desk-Expert 12 software (Table 4). The water content of the DES, extraction time, extraction temperature, and solid-liquid ratio were selected as the main influencing factors. Three levels, namely, low, medium, and high were designed for each factor and marked −1, 0, and 1, respectively.

### 4.5. In Vitro Antioxidant Activity of S. baumii Polyphenols

The deep eutectic solvent extract was purified by macroporous resin(AB-8) to obtain crude polyphenols. Then the obtained crude polyphenols were prepared into sample solution with concentrations of 0.10, 0.15, 0.20, 0.25, and 0.30 mg/mL, respectively, for reserve.

#### 4.5.1. •OH-Scavenging Activity of S. baumii Polyphenols

The scavenging ability of polyphenols on •OH was determined using the H_2_O_2_/Fe method, following the method reported by Smirnoff [31] with slight modifications. VC was used as a control, and 9 mmol/L FeSO_4_ and 9 mmol/L salicylic acid-ethanol were prepared for later use. A volume of 250 µL of 30% H_2_O_2_ was diluted to 250 mL with water and stored away from light in an amber reagent bottle until used.

In a test tube, 1 mL FeSO_4_, 2 mL salicylic acid-ethanol solution, and 2 mL distilled water were mixed, and 2 mL H_2_O_2_ solution was added and quickly mixed. The sample was incubated for 1 h in a water bath at 37 ℃, and the absorbance was measured at 510 nm to obtain the value for the blank control group, A_0_.

Next, 1 mL FeSO_4_, 2 mL salicylic acid-ethanol solution, and 2 mL of distilled water, were added to 1 mL of the extract obtained using different concentration gradients (0.10, 0.15, 0.20, 0.25, and 0.30 mg/mL). After mixing, the samples were incubated in a water bath at 37 ℃ for 1 h. The absorbance was measured at 510 nm and readings for the blank reagent group were obtained, which were labeled as A_y_.

Lastly, 1 mL of the extract obtained using different concentration gradients (0.10, 0.15, 0.20, 0.25, and 0.30 mg/mL) was added to 1 mL FeSO_4_ and 2 mL salicylic acid-ethanol solution in the respective test tubes. After mixing, 2 mL of H_2_O_2_ was added. After rapid mixing, the samples were incubated in a water bath at 37 ℃ for 1 h, and the absorbance was determined at 510 nm to obtain readings for the sample group, marked as A_x_. The Scavenging rate was calculated using the following Equation (2):(2)$$\mathrm{Scavenging}\;\mathrm{rate}\;\left(\%\right)=\frac{A_{0}-\left(A_{x}{-A}_{y}\right)}{A_{0}}\text{×}100\%$$

#### 4.5.2. DPPH-Scavenging Activity of S. baumii Polyphenols

The DPPH-scavenging ability of polyphenols was determined using the method reported by Xican Li [32] with slight modifications. VC was used as the control. A 0.004% DPPH-ethanol solution was prepared and stored in a refrigerator at 4 ℃ until used. Sample solutions of different concentrations (0.10, 0.15, 0.20, 0.25, and 0.30 mg/mL) were added into the respective test tubes, followed by the addition of 1 mL of 0.0004% DPPH solution and 2 mL of anhydrous ethanol. After thorough mixing, the reaction was allowed to progress by keeping the test tubes undisturbed in the dark for 30 min. The absorbance was measured at 517 nm and marked as A_x_. Distilled water was used as a blank and marked as A_0_. The blank control group was set as a sample, which was marked as A_y_. The Scavenging rate was calculated using the following Equation (3):(3)$$\mathrm{Scavenging}\;\mathrm{rate}\left(\%\right)=\frac{A_{0}-\left(A_{x}{-A}_{y}\right)}{A_{0}}\text{×}100\%$$

#### 4.5.3. ABTS^+^-Scavenging Activity of S. baumii Polyphenols 

The ABTS^+^-scavenging ability of polyphenols was determined using the method reported by Miller [33] with slight modifications. VC was used as a control. ABTS^+^ (7.4 mmol/L) was mixed with 2.6 mmol/L K_2_S_2_O_8_ in a 1:1 ratio, and the reaction was allowed to progress in the dark at 25 ℃ for 12 h. After a 40–50-fold dilution, the absorbance was measured at 734 nm. Sample solutions of different concentrations (0.10, 0.15, 0.20, 0.25, and 0.30 mg/mL) were added to 4 mL of ABTS^+^ working solution, mixed thoroughly, and allowed to stand for 6 min. The absorbance was determined at 734 nm and marked as A. Anhydrous ethanol was used as a blank control and marked as A_0_. The Scavenging rate was calculated using the following Equation (4):(4)$$\mathrm{Scavenging}\;\mathrm{rate}\left(\%\right)=\frac{A_{0}-A}{A_{0}}\times 100\%$$

## 5. Conclusions

In this study, we have reported a novel method for the extraction of polyphenols from *S. baumii* using DES. In the single-factor experiment, using the same extraction conditions (solid–liquid ratio of 50 mg/mL, ultrasonic-assisted extraction for 30 min, temperature of 50 ℃, power of 200 W, 40kHz), we compared the results obtained using ethanol extraction (60% ethanol was found to be the optimal extraction condition). When DES was used for extraction, the polyphenol yield was 6.37 mg/g(DES-1), which was 1.5 times more than that obtained using 60% ethanol (4.20 mg/g). These findings preliminarily indicate the extraction yield of polyphenols using a eutectic solvent. Based on our initial findings, we used response surface analysis to further enhance the extraction conditions using DES. The ideal extraction time was 42 min, the extraction temperature was 58 ℃, the solid–liquid ratio was 34 mg/mL, and the water content was 39%. Using these conditions, the polyphenol yield from *S. baumii* was found to be 12.58 mg/g, which was about 2 times as much as before conditional optimization (6.37mg/g). This result demonstrated the superiority of DES to extract polyphenols. When the concentration of the polyphenols was 0.30 mg/mL, the •OH-, DPPH-, and ABTS^+^- scavenging rates were 95.71%, 91.08%, and 85.52%, respectively. These findings indicated that *S. baumii* polyphenols were potent-free radical scavengers and natural antioxidants, further highlighting their broad application prospects in drug development and the research and development of health supplements.

## Figures and Tables

**Figure 1 molecules-26-03850-f001:**
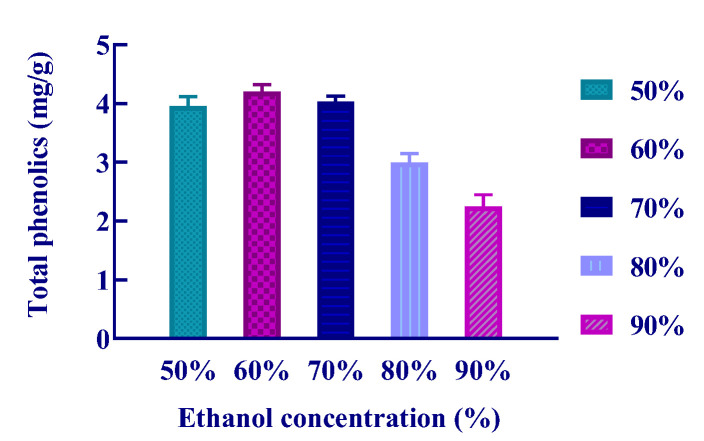
Effect of ethanol concentration on the extraction yield of *S. baumii* polyphenols.

**Figure 2 molecules-26-03850-f002:**
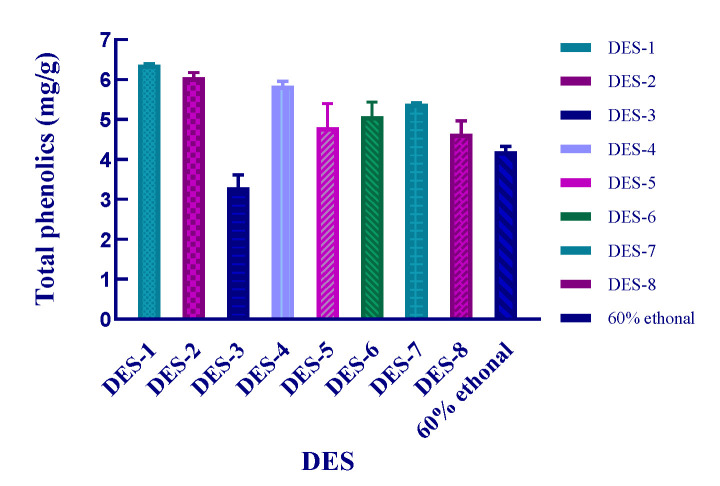
Effects of different DES on the extraction yield of polyphenols in *S. baumii*.

**Figure 3 molecules-26-03850-f003:**
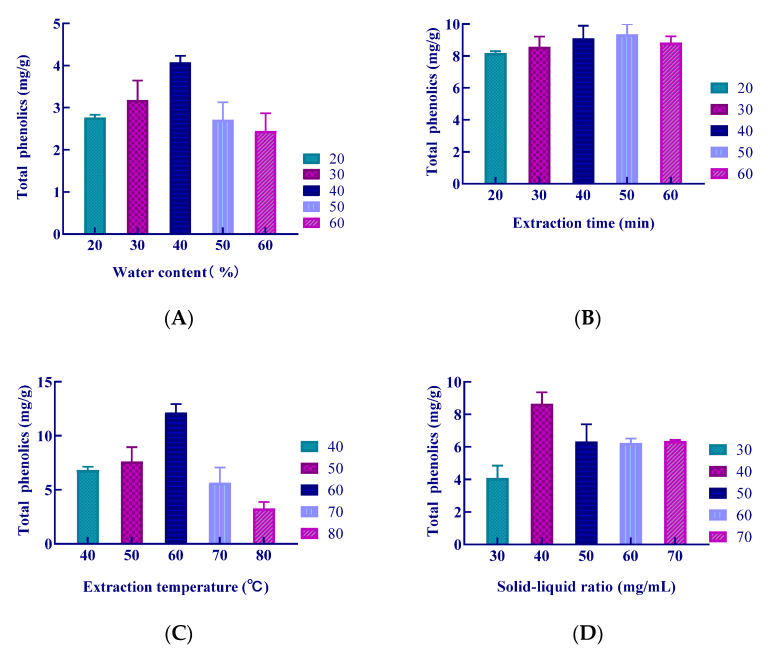
Effect of different extraction conditions on the extraction yield of polyphenols.

**Figure 4 molecules-26-03850-f004:**
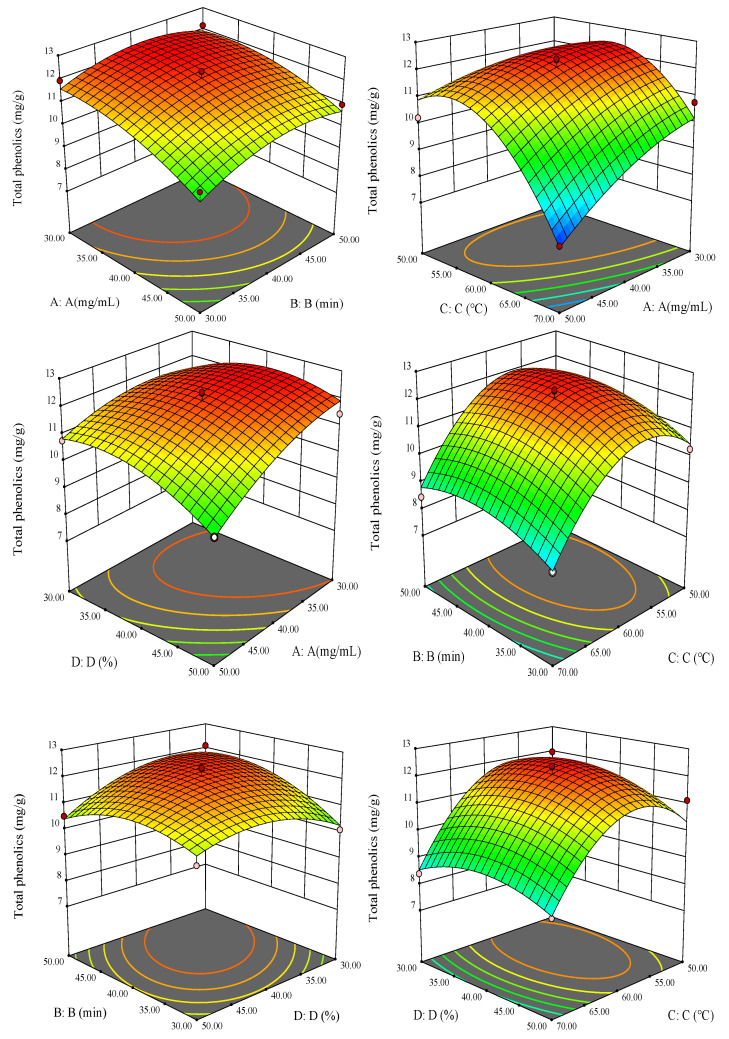
Response surface of the effects of the interaction of various factors on the extraction yield of *S. baumii* polyphenols.

**Figure 5 molecules-26-03850-f005:**
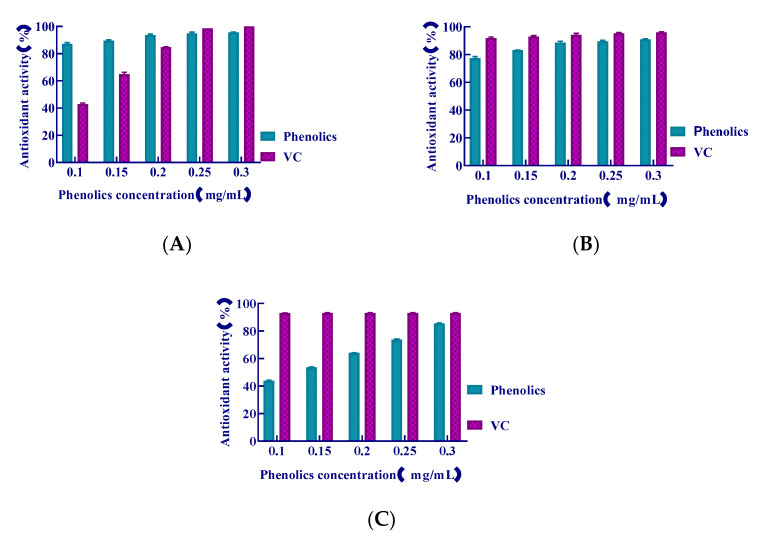
The experimental results of antioxidant activity of polyphenols.

**Table 1 molecules-26-03850-t001:** Response surface experimental design scheme and results.

Number	Ratio of the Material Liquid(mg/mL)A	Extraction Time(min)B	Extraction Temperature (°C)C	The Water Content(%)D	Polyphenols Content(mg/g)
1	40	40	70	30	8.39
2	40	30	60	50	10.39
3	40	40	60	40	12.37
4	40	30	50	40	10.27
5	30	40	60	50	11.43
6	40	40	50	30	11.97
7	40	50	60	30	12.09
8	30	40	60	30	10.96
9	50	40	60	30	10.77
10	30	50	60	40	12.17
11	40	30	70	40	8.27
12	40	40	60	40	12.33
13	30	40	70	40	10.74
14	50	40	70	40	7.32
15	40	40	60	40	11.92
16	30	30	60	40	11.93
17	40	40	50	50	11.12
18	40	50	50	40	10.68
19	40	50	60	50	10.51
20	50	40	60	50	9.53
21	40	40	60	40	11.73
22	40	30	60	30	10.09
23	50	50	60	40	10.96
24	40	40	60	40	12.03
25	50	30	60	40	10.14
26	30	40	50	40	10.98
27	40	50	70	40	8.45
28	40	40	70	50	8.44
29	50	40	50	40	10.22

**Table 2 molecules-26-03850-t002:** Variance analysis of extraction yield using the regression model.

Sources Model	Sum of Squares	DF	Mean Square	*F*-Value	*p*-Value	
	49.320	14	3.520	12.720	<0.0001	Significant
A-A	7.160	1	7.160	25.860	0.0002	
B-B	1.180	1	1.180	4.280	0.0576	
C-C	15.480	1	15.480	55.900	<0.0001	
D-D	0.687	1	0.677	2.440	0.1403	
AB	0.084	1	0.084	0.304	0.5903	
AC	1.770	1	1.770	6.390	0.0242	
AD	0.731	1	0.731	2.640	0.1265	
BC	0.013	1	0.013	0.048	0.8302	
BD	0.884	1	0.884	3.190	0.0957	
CD	0.202	1	0.203	0.731	0.4069	
A^2^	1.430	1	1.430	5.180	0.0392	
B^2^	2.490	1	2.490	9.010	0.0095	
C^2^	20.040	1	20.040	72.350	< 0.0001	
D^2^	2.760	1	2.760	9.970	0.0070	
Residual	3.880	14	0.277			
Lack of fit	3.580	10	0.358	4.820	0.0716	Not Significant
Pure Error	0.297	4	0.074			
R^2^	0.927					

**Table 3 molecules-26-03850-t003:** Composition of DES.

Number	Composition	Molar Ratio
DES-1	Choline chloride: Malic acid	1:1
DES-2	Choline chloride: Lactic acid	1:1
DES-3	Choline chloride: Glycerol	1:2
DES-4	Choline chloride: Malonic acid	1:1
DES-5	Choline chloride: Urea	1:2
DES-6	Choline chloride: Oxalic acid	1:1
DES-7	Choline chloride: 1,4-Butanediol	1:5
DES-8	Choline chloride: Urea: Glycerol	1:1:1

**Table 4 molecules-26-03850-t004:** Factors and levels in response surface analysis.

Factors	Level		
	−1	0	1
A ratio of the material liquid (mg/mL)	30	40	50
B Extraction time (min)	30	40	50
C Extraction temperature (℃)	50	60	70
D The water content (%)	30	40	50

## Data Availability

All the data is shown in the article, andthe study did not report any data.

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
