# Peer review of "Optimization of Extraction Process and the Antioxidant Activity of Phenolics from Sanghuangporus baumii"

_molecules, 2021, doi:10.3390/molecules26133850_

Round 1

Reviewer 1 Report

I found the paper very interesting. It was written very clearly, but I would like to address some comments:

  1. The abstract is too long. It should be a total of about 200 words maximum.
  2. Figures 2, 3, and 5. The text on the chart's axis is illegible.
  3. Line 134, 138, 191, 228: S. baumii should be written in italic.
  4. Line 135: It should be writing: Figure 3 (A).
  5. Line 171-173: Symbol of statistical parameters should be written in italic.
  6. The discussion is too short. The authors should discuss the results from the perspective of previous studies.
  7. Line 301: Equation: The symbol for mass is m, not M.
  8. Line 302-303: Symbols for physical quantities should be writing italic.
  9. Line 323: Insert space between 50 mg/ml.
  10. What does the mean letter “J” in references?

Author Response

Dear Editor and reviewer,

We have studied the valuable comments from you, the assistant editor and reviewers carefully, and tried our best to revise the manuscript. The revised manuscript (The original No. 1249549) had been uploaded.

We appreciate for Editors/Reviewers’ warm work earnestly, and hope that the correction will meet with approval. Once again, thank you very much for your comments and suggestions.

I am looking forward to hearing from you.

Sincerely yours,

Rui Zhang

Reviewer 2 Report

Manuscript Number: MDPI

Title: Optimization of Ultrasonic-Assisted Deep Eutectic Solvent Extraction and the Antioxidant Activity of Phenolic Compounds from Sanghuang

Comments

The paper deals with the effects of Deep Eutectic Solvent (DES) and operating parameters on the ultrasound-assisted extraction (UAE) of polyphenols from Sanghuang and the antioxidativity. Based on the results from single-factor experiments, response surface methodology was used to optimize the extraction conditions. Under optimal conditions, the highest extraction efficiency of polyphenols (12.58 mg/g) was achieved. The extracted polyphenols exert a certain antioxidant activity by in vitro tests. Unfortunately, this manuscript was roughly prepared and the discussion on the obtained results is very limited. No theoretical explanations were found for almost all the results. In particular, the influence of different EDS on extraction efficiency should be discussed in details. A major revision is required for re-submission. The main comments are listed as follows:

  1. The title and keywords of the article need to be revised to conform to the general concept.
  2. The diagram of the UAE device or process scheme, as well as the important extraction parameters, such as ultrasonic frequency, ultrasonic power intensity or density, solid particle size, total extraction volume, etc., need to be shown or marked.
  3. How did you purify or separate polyphenols from extraction matrice? If the purification or separation was not performed, how did you carried out the antioxidative test?
  4. The terms used in this article is random and confusing, as shown in follows:
  • Phenolic compounds, Phenolic compound, phenolic constituents, phenolics, phenolic substances, polyphenol, polyphenols, phenol, etc;
  • Ultrasonic-assisted extraction, ultrasound-assisted extraction, ultrasonic extraction;
  • The free-radical scavenging ability of the phenolic compound, the ability of the phenolics to scavenge; The free-radical scavenging rate, The clearance rate;
  • extraction efficiency, extraction rate, extraction yield, phenol content, polyphenol content;
  • extraction time, extracting time;
  • water content, moisture content, water level;

Terminology should be used uniformly in accordance with general scientific concepts.

  1. In Fig. 1, Fig.2, Fig. 3, legends are redundant and should be deleted. It is hard to understand the meaning of “phenol content” on the Y-axis, “percent ethanol” on the X-axis. In Fig.4, A:A, B:B, C:C, D:D and the term “phenol content” cannot be understand. In Fig. 5, , the term “phenol content” and “scavenging rate” cannot be understand.
  2. In Table 1, an important factor - extraction temperature (C) cannot be found.
  3. All calculation equations should be numbered. Eq. 2 and Eq. 3 seem to be the same.
  4. In the section of 3. Discussion, nothing was discussed. It looks like future works.
  5. In the conclusion, in the following two sentences, which multiple expression is correct?
  • When DES were used for extraction, the polyphenol yield was 6.37 mg/g, which was 1.5 times more than that obtained using 60% ethanol (4.20 mg/g).
  • Using these conditions, the polyphenol yield from S. baumii was found to be 12.58 mg/g, which was 3 times that obtained using 60% ethanol (4.20 mg/g) in the single factor experiment
  1. The following sentences may be incorrect or confusing and need to be revised.
  • research on the polyphenols of Sanghuang is mainly focused on polysaccharides and flavonoids, and very few studies have studied the polyphenols present in this fungus.
  • Other important extraction methods include ultrasound-assisted extraction[20], microwave-assisted extraction[21], enzyme-assisted extraction[22], supercritical extraction[23], and subcritical water extraction[24]. These traditional methods have some disadvantages, such as extended extraction time, low purity of the extracted components, low extraction rate, and the possibility of environmental pollution. (If so, it is difficult to understand the motivation of using ultrasound-assisted extraction in this work.)
  • At the same time, different volumes and concentrations of ethanol were selected as controls to more directly reflect the differences between the two extraction systems. (The effect of ethanol volumes cannot be found.)
  • The experimental results show that the polyphenol yield increased first and then decreased with an increase in water content. When the water content was 40%, the yield of the extracted polyphenols reached the maximum, and then decreased as the water level increased. (These are very long-winded expression and need to be simplified.)
  • The ABTS+-scavenging rate was the highest at 85.5% when the phenol concentration was 0.3 mg/mL, and was 91.82% higher than that of VC. 

Author Response

(The authors gave the same response as above.)
